# Negative health impacts of navigating the healthcare system for musculoskeletal conditions: A scoping review protocol

Geneviève Jessiman-Perreault[1], Paige Campbell[2], Dawn Henley[3], Danika Tribo[4], Ania Kania-Richmond[1], Breda H. F. Eubank[1,3]*

1 Bone and Joint Health Strategic Clinical Network, Alberta Health Services, Edmonton, Alberta, Canada, 2 Cumming School of Medicine, University of Calgary, Calgary, Alberta, Canada, 3 Faculty of Health, Community, & Education, Mount Royal University, Calgary, Alberta, Canada, 4 Surgery Strategic Clinical Network, Alberta Health Services, Edmonton, Alberta, Canada

* beubank@mtroyal.ca

**Data Availability Statement:** All relevant data (i.e. search strategy) are within the paper.

## Abstract

Musculoskeletal (MSK) conditions, particularly shoulders, knees, and the low back issues, place a significant burden on individuals, society, and healthcare systems. There is a lack of attention to negative health effects impacting patients because of their interactions to access appropriate diagnostics, assessments, and treatments. This scoping review intends to search and synthesize peer-reviewed evidence on the negative health impacts associated with navigating the healthcare system for MSK care. A scoping review will be conducted following the PRISMA guidelines for Scoping Reviews and Arksey and O'Malley's 5-step process. Six databases will be searched with no time or geographic limits. Included articles must meet all the following criteria: 1) the patients must be adults, 2) patients must be seeking care for their knee, low-back, or shoulder condition, 3) interacted with the healthcare system, and 4) experienced health impacts due to navigating the healthcare system. Information from each article will be charted in a pre-determined extraction. This protocol aims to share our methods ahead of analysis to increase rigour and transparency. The scoping review results will better elucidate the health impacts of the inaccessibility of high-quality care for MSK conditions. The findings also aim to inform the development of patient-centered outcomes to evaluate alterations to the current MSK pathways.

## Introduction

Musculoskeletal (MSK) conditions, particularly those affecting the low back, knees, and/or shoulders, are a substantial contributor to the global burden of disease [1] and pose a significant burden to individual dysfunction and disability, and healthcare utilization [2–5]. These conditions have substantial impacts on individual quality of life and for effective management of symptoms and dysfunction, require multiple interactions with the healthcare system to assess, diagnose and treat these conditions. The prevalence of MSK conditions have been steadily increasing in recent years [6] and it is expected that they will continue to do so [7,8]

**Funding:** The author(s) received no specific funding for this work.

**Competing interests:** The authors have declared that no competing interests exist.

due to factors such as obesity and an aging population [9,10], increasing demand on health care systems.

Globally, many healthcare systems struggle with this increasing demand, and this has contributed to longer wait times for MSK assessment and specialist treatments such as knee and hip surgery [11,12]. This trend has worsened due to delays resulting from the COVID-19 pandemic [13–16] and shortages of primary healthcare providers [17] and providers with specialized training to assess, diagnose, and treat MSK conditions [18–20]. Moreover, many patients struggle to navigate the healthcare system to receive appropriate diagnosis and care for their MSK condition due to barriers around access to care, logistical issues, cost of treatment, lack of familiarity with providers, poor communication, lack of follow up, limited access to non-surgical care, and lengthy wait times for services [21–23].

These barriers to obtaining care for MSK conditions have resulted in poorer outcomes for patients as their health often deteriorates during an extended period of waiting [14,24,25]. However, less is known about the negative impacts on the various dimensions of health associated with health system interactions when seeking services to address MSK conditions. This scoping review aims to fill this gap by searching and synthesizing peer-reviewed literature to understand the health impacts associated with navigating the healthcare system when seeking care for MSK conditions.

## Material and methods

### Study design

A scoping review will be conducted following the Arksey and O'Malley's 5-step process: (1) identifying the research question; (2) identifying relevant studies; (3) study selection; (4) charting the data; and (5) collating, summarizing, and reporting the results [26]. The reporting of this scoping review protocol is in accordance with the Preferred Reporting Items for Systematic review and Meta-Analysis Protocols (PRISMA-P), the completed PRISMA-P checklist is provided in S1 Appendix. The results of this scoping review will be reported according to the Preferred Reporting Items for Systematic Reviews and Meta-Analysis (PRISMA) extension for Scoping Reviews (PRISMA-ScR) reporting guide [27]. The PRISMA-ScR guideline was used because it is widely used by researchers and considered best practice for rigorous and transparent approach to literature searches.

### Eligibility criteria

This study protocol will be applied to understand the health impacts of navigating the healthcare system for low back, knee, or shoulder problems by drawing on peer reviewed publications. The exclusion and inclusion criteria have been developed to better understand health impacts from a holistic approach and to understand what is meant by navigating the healthcare system.

**Inclusion criteria.** All included articles must meet the following four criteria: 1) the patient must be an adult (i.e., 18 years of age or older), 2) the patient must be seeking care for a knee, low-back, and/or shoulder condition, 3) the patient must have interacted with the healthcare system (i.e., acute, primary care, community health) or attempted to do so, and 4) the patient must have experienced health impacts due to experience navigating the healthcare system.

Models of holistic health are highly variable and include upwards of 379 domains clustered under 14 themes [28]. For this scoping review, the health impact described in the included paper must fit within one or more of the eight dimension model of health which includes physical, social, emotional, intellectual, spiritual, occupational, environmental, and psychological health [29,30]. These eight dimensions of health are described, and examples are provided of each in Table 1 below.

**Table 1. Definitions and examples of eight dimensions of health adapted from Oliver et al [30].**

| Dimension | Definition | Example |
|---|---|---|
| Physical | This dimension refers to the proper functioning of the body and its systems. | Having a balanced diet, regular exercise, adequate sleep, and the absence of chronic diseases, disabilities, or pain. |
| Social | This dimension relates to the ability to form and maintain meaningful relationships with others and the community. | Having a supportive network of family and friends, effective communication skills, and the ability to adapt to different social situations. |
| Emotional | This dimension involves the ability to express and manage emotions in a healthy way. | Having healthy coping mechanisms, maintaining optimism, and developing emotional intelligence. |
| Intellectual | This dimension refers to the ability to engage in creative and stimulating mental activities | Continuing education, problem-solving skills, and engaging in intellectually stimulating hobbies or activities |
| Spiritual | This dimension involves finding meaning and purpose in life and developing a sense of inner peace and well-being. | Practicing meditation, engaging in religious or spiritual practices, and exploring personal values and beliefs. |
| Occupational | This dimension relates to the satisfaction and fulfillment derived from one's work or occupation and ability to balance ones work responsibilities with other demands. | Finding a balance between work and personal life, feeling a sense of accomplishment and purpose in one's work, and maintaining a safe and healthy work environment. |
| Environmental | This dimension encompasses the recognition of the impact of the environment on overall health and well-being. | Living in a safe and clean environment, practicing sustainable living, and being aware of environmental hazards. |
| Psychological/ Mental | This dimension encompasses emotional, psychological, and cognitive well-being. | Managing stress effectively, having a positive self-image, and the ability to cope with life's challenges. |

**Exclusion criteria.** Articles are excluded if any of the four criteria apply: 1) health impact is related to treatment of the condition through medical intervention rather than navigating the healthcare system, 2) full-text article is not available, 3) article is not available in English, and 4) article is a protocol (i.e., does not report on results).

## Information source

This scoping review will include peer reviewed publications from database inception to the date the search is conducted (i.e., no time limits will be applied). In addition, no limits will be applied based on geography, study time, or study type (i.e., qualitative, quantitative, and mixed methods papers will be included). A language-based limit will be applied to only include articles published in or translated into English.

## Search strategy

The search was designed and conducted with the expert guidance from a research librarian. This process began with the development of the search question (What are the negative impacts of navigating healthcare for musculoskeletal conditions?) and search concepts. Thus, the search strategy was based on 3 main concepts: 1) chronic or acute musculoskeletal conditions (low back, shoulder, knees) including generalized MSK pain in those areas, 2) healthcare interactions, 3) health outcomes. One additional search concept was included to limit articles to the adult population. Six databases (APA PsychInfo, CINAHL, Cochrane Library, Medline (OVID), Psychology & Behavioral Sciences Collection, PubMed) were searched, these databases were selected based on their applicability to health sciences, social sciences, and medicine. Exemplary papers (i.e., papers that fulfilled the inclusion criteria) were selected at the outset of the search; after conducting the search we scanned the results to ensure that those papers were captured by the search strategy. The search strategy was revised once in consultation with the research team to broaden the terms related to navigation. The search strategy used for each database is presented in Table 2.

**Table 2. Databases and associated search strategy.**

| Database Name | Search Strategy |
| --- | --- |
| APA PsycInfo | 1 exp Patients/ or patients.mp.<br>2 Older Adulthood/ or Late Adulthood/ or Middle Adulthood/<br>3 (adult or adults).mp.<br>4 2 or 3<br>5 1 and 4<br>6 exp "Back (Anatomy)"/ or back.mp.<br>7 knee.mp. or exp Knee/<br>8 exp "Shoulder (Anatomy)"/ or shoulder.mp.<br>9 6 or 7 or 8<br>10 exp Health Care Delivery/ or "delivery of healthcare".mp.<br>11 "patient care".mp.<br>12 treatment.mp. or exp Treatment/<br>13 exp Diagnosis/ or diagnosis.mp.<br>14 10 or 11 or 12 or 13<br>15 "patient satisfaction".mp. or exp Client Satisfaction/<br>16 "patient experience".mp.<br>17 "patient perspect*".mp.<br>18 "patient percept*".mp.<br>19 15 or 16 or 17 or 18<br>20 5 and 9 and 14 and 19<br>21 limit 20 to english language |
| CINAHL | S12 S3 AND S4 AND S9 AND S10 (with English language filter)<br>S11 S3 AND S4 AND S9 AND S10<br>S10 (MH "Patient Attitudes") OR "Patient satisf' or patient experience or patient perspective' or patient perception´´´ OR (MH "Patient Preference")<br>S9 S5 OR S6 OR S7 OR S8<br>S8 "patient care or treat´´´ OR (MH "Patient Care+")<br>S7 (MH "Diagnosis+") OR "diagnos´´´ OR (MH "Diagnosis, Delayed")<br>S6 (MH "Outcome Assessment") OR (MH "Pain Measurement") OR "assess' or measure´´´ OR (MH "Patient Assessment+") OR (MH "Physical Therapy Assessment") OR (MH "Occupational Therapy Assessment") OR (MH "Needs Assessment") OR (MH "Quality Assessment+") OR (MH "Self Assessment")<br>S5 (MH "Health Care Delivery+") OR "health care system' or health care deliver' or health care service' or healthcare system' or healthcare deliver' or healthcare service' or health service´´´ OR (MH "Tertiary Health Care") OR (MH "Secondary Health Care") OR (MH "Medical Care+") OR (MH "Health Care Industry") OR (MH "Primary Health Care") OR (MH "Health Facilities+")<br>S4 ((MH "Back") OR "back or lumbar or low back") OR ((MH "Knee") OR (MH "Knee Joint+") OR "knee or knees or tibiofibular´´´) OR ("shoulder´´´ OR (MH "Shoulder Joint+") OR (MH "Shoulder"))<br>S3 S1 AND S2<br>S2 (MH "Health Care Delivery+") OR "health care system' or health care deliver' or health care service' or healthcare system' or healthcare deliver' or healthcare service' or health service´´´ OR (MH "Tertiary Health Care") OR (MH "Secondary Health Care") OR (MH "Medical Care+") OR (MH "Health Care Industry") OR (MH "Primary Health Care") OR (MH "Health Facilities+")<br>S1 "patient or patients or client or clients" OR (MH "Patients+") |
| Cochrane Library | S1 ((patient* or client* or inpatient* or outpatient*)):ti,ab,kw AND ((adult or adults or aged or elder* or senior* or middle NEXT age or middle NEXT aged)):ti,ab,kw<br>S2 ((back or lumbar or low NEXT back)):ti,ab,kw OR ((knee or knees)):ti,ab,kw OR (shoulder*):ti,ab,kw<br>S3 ((health NEXT care NEXT system* or health NEXT care NEXT deliver* or health NEXT care NEXT service* or healthcare NEXT system* or healthcare NEXT deliver* or healthcare NEXT service* or health NEXT service*)):ti,ab,kw OR ((assess* or measure*)):ti,ab,kw OR ((diagnos*)):ti,ab,kw OR ((patient NEXT care or treat*)):ti,ab,kw<br>S4 ((Patient NEXT satisf* or patient NEXT experience* or patient NEXT perspective* or patient NEXT perception*)):ti,ab,kw<br>S5 #1 AND #2 AND #3 AND #4 |

(*Continued*)

**Table 2.** (Continued)

| Database Name | Search Strategy |
|---|---|
| Medline (OVID) | 1 patients.mp. or exp Patients/<br>2 exp Adult/ or adult.mp.<br>3 1 and 2<br>4 exp Back Muscles/ or exp Back/ or back.mp.<br>5 exp Knee Joint/ or "knee joint".mp.<br>6 "shoulder joint".mp. or exp Shoulder Joint/<br>7 4 or 5 or 6<br>8 exp "Delivery of Health Care"/ or "delivery of healthcare".mp.<br>9 "patient care".mp. or exp Patient Care/<br>10 exp Diagnosis/ or diagnosis.mp.<br>11 8 or 9 or 10<br>12 exp Patient Satisfaction/ or "patient satisfaction".mp.<br>13 "patient experience".mp.<br>14 "patient perspect*".mp.<br>15 "patient percept*".mp.<br>16 12 or 13 or 14 or 15<br>17 3 and 7 and 11 and 16<br>18 limit 17 to english language |
| Psychology & Behavioral Sciences Collection | S6 S1 AND S2 AND S3 AND S4 (with English language filter)<br>S5 S1 AND S2 AND S3 AND S4<br>S4 "Patient satisf*" or "patient experience*" or "patient perspective*" or "patient perception*"<br>S3 ("health care system*" or "health care deliver*" or "health care service*" or "healthcare system*" or "healthcare deliver*" or "healthcare service*" or "health service*") OR (assess* or measure*) OR diagnos* OR ("patient care" or treat*)<br>S2 (back or lumbar or "low back") OR (knee or knees) OR shoulder*<br>S1 (patient* or client* or inpatient* or outpatient*) AND (adult or adults or aged or elder* or senior* or "middle age" or "middle aged") |
| PubMed | S6 #1 AND #2 AND #3 AND #4 (with English language filter)<br>S5 #1 AND #2 AND #3 AND #4<br>S4 "patient satisf*"[Text Word] OR "patient experience*"[Text Word] OR "patient perspective*"[Text Word] OR "patient perception*"[Text Word]<br>S3 "health care system*"[Text Word] OR "health care deliver*"[Text Word] OR "health care service*"[Text Word] OR "healthcare system*"[Text Word] OR "healthcare deliver*"[Text Word] OR "healthcare service*"[Text Word] OR "health service*"[All Fields] OR "assess*"[All Fields] OR "measur*"[All Fields] OR "diagnos*"[All Fields] OR "patient care"[Text Word] OR "treat*"[Text Word]<br>S2 "back"[Text Word] OR "lumbar"[Text Word] OR "low back"[Text Word] OR "knee"[Text Word] OR "knees"[Text Word] OR "shoulder*"[Text Word]<br>S1 ((patient*[Text Word] OR client*[Text Word] OR inpatient*[Text Word] OR outpatient*[Text Word]) AND (adult[Text Word] OR adults[Text Word] OR aged[Text Word] OR elder*[Text Word] OR senior*[Text Word] OR "middle age"[Text Word] OR "middle aged"[Text Word])) |

## Study records

**Data management.** Covidence software will be used to remove duplicates and facilitate screening among researchers [31]. Once study selection is complete, the included articles will be extracted from Covidence for further analysis.

**Selection process.** Prior to the beginning of the selection process, a sub-set of articles (n = 10) will be reviewed independently by all reviewers for inclusion. The reviewers will then meet to discuss any discrepancies and make any necessary alterations to the inclusion or exclusion criteria. Once this pilot testing is complete, the title and abstract screening will begin. This first stage of screening involves four independent reviewers (GJP, DH, PC, DT) reading the titles and abstracts of each article and determining inclusion or exclusion based on the pre-defined eligibility criteria. Articles are included or excluded based on a vote from two reviewers. Any discrepancies will be resolved by consulting with the senior researchers on the team

(BHFE, AKR) in the authorship team. In the second stage of screening, the articles marked for full-text review will be read in their entirety by the screeners and will be assessed for inclusion by two independent reviewers. Any discrepancies in screening will be resolved in consultation with the senior researchers on the team (BHFE, AKR). The studies selected for inclusion will undergo an additional screening process called 'backwards citation chaining.' First, we will review the reference lists of the included studies to identify any potentially relevant sources cited within them; and second, checking other resources that have cited the included studies to determine if any references meeting the inclusion criteria were initially missed. The citation chaining process will be conducted using Web of Science All articles included in the final synthesis will be agreed upon by the entire authorship team.

**Data collection.** After final articles are determined, data will be extracted from articles by one researcher. Data will be charted in an extraction table that has been agreed upon by the authorship team a priori (see Table 3).

## Data items and charting process

One reviewer will lead the data charting and synthesis, which will be reviewed by all members of the authorship team to ensure accuracy and completeness of extracted information. The following data will be charted: title, authors, year, journal, study design, sample size, framework, data collection instrument, study setting, location of study, type of healthcare system, type of condition, co-morbidities, type of interaction with the healthcare system, overall study results (i.e., patient experience), health impact (social, emotional, spiritual, intellectual, physical, environmental, psychological, and occupational health), and recommendations for quality improvement.

## Outcomes and prioritization

The findings will be narratively synthesized to form the results of the study and provide details on the range of negative health impacts experienced by patients when seeking MSK health care services, what aspects of navigation were related to these health outcomes, and any recommendations to improvement to the MSK care system or additional patient supports that may be necessary to reduce these negative outcomes resulting from interactions with a health care system. In addition, given that we have not placed any geographic limits on this scoping review,

**Table 3. Extraction template.**

| Article Descriptors | | | | | Methodology | | | | Description of Setting | | | Results and Conclusions | | | | | |
|---|---|---|---|---|---|---|---|---|---|---|---|---|---|---|---|---|---|
| Article ID | Title | Author (s) | Year | Journal | Study Design | Sample Size | Framework/ Theory | Data Collection Instrument | Study Setting (e.g., acute care, primary care, community health) | Location of Study (e.g., country or region) | Type of Healthcare System (e.g., universal healthcare system, social health insurance, private health insurance, out-of-pocket system, mixed system, community-based or faith-based healthcare systems) | Overall Study results | Type of MSK Condition | Co-morbidities or other conditions | Type of Interaction with Healthcare | Health Impact | Recommendations |
| 101 | | | | | | | | | | | | | | | | | |
| . . . | | | | | | | | | | | | | | | | | |

we will examine whether there are any differences in negative health impacts, or causes of impacts, based on the type of healthcare system in place in the study setting.

## Discussion

The need for essential, high-quality, and timely care MSK care is increasing, particularly in high and medium-income countries at a time when these healthcare systems are experiencing resource constraints to provide care [32,33]. Therefore, patients with MSK care may be suffering from not only physical deterioration but also other holistic health impacts due to their negative experiences of navigating an increasingly complex and inaccessible system. It is crucial to identify these negative health impacts to better understand the full impact of the current state of the MSK healthcare system from a patient perspective. We believe that our findings will be helpful for health professionals who are aiming to improve MSK health care systems as these results will provide a more fulsome understanding of the adverse health outcomes associated with the current system. Moreover, we believe these findings can be used to develop patient-centered evaluation measures to better evaluate the success of future alterations to the current MSK care pathways.

### Limitations

The exclusion of non-English language papers may limit our scoping review, which aims to have an international perspective. However, we expect the impact to be minimal given most peer-reviewed articles are published in or translated to English. Our search is not expected to yield results for all MSK conditions (e.g., hip, neck, or hand), instead we narrowed our scope to only search for papers that examine the most common MSK conditions (i.e., low back, knee, shoulders) [6]. We also anticipate that only searching peer-reviewed publications could result in publication bias, the phenomenon where researchers are more likely to publish positive results. Given that we are searching for negative health outcomes associated with healthcare navigation, it is possible that expanding our search to grey literature could yield more results.

## Conclusion

The results from this proposed scoping review should provide health professionals with a full scope of patient centered outcomes of the current MSK healthcare system. This information will provide valuable information to inform quality improvement interventions and inform more robust evaluation metrics.

## Supporting information

**S1 Appendix. PRISMA-P (Preferred Reporting Items for Systematic review and Meta-Analysis Protocols) 2015 checklist: recommended items to address in a systematic review protocol\*.**
(DOCX)

## Acknowledgments

We would like to express our gratitude to Marissa Rocca from the Knowledge Resource Service at Alberta Health Services for her valuable contribution to the development of the search strategy for this scoping review.

## Author Contributions

**Conceptualization:** Geneviève Jessiman-Perreault, Paige Campbell, Dawn Henley, Danika Tribo, Ania Kania-Richmond.

**Methodology:** Geneviève Jessiman-Perreault, Paige Campbell, Dawn Henley, Danika Tribo, Ania Kania-Richmond, Breda H. F. Eubank.

**Project administration:** Geneviève Jessiman-Perreault, Ania Kania-Richmond.

**Resources:** Ania Kania-Richmond.

**Supervision:** Geneviève Jessiman-Perreault, Breda H. F. Eubank.

**Writing – original draft:** Geneviève Jessiman-Perreault.

**Writing – review & editing:** Geneviève Jessiman-Perreault, Paige Campbell, Dawn Henley, Danika Tribo, Ania Kania-Richmond, Breda H. F. Eubank.

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
