## [Decision Letter · Decision Letter 0]

15 Aug 2024

PONE-D-24-26506Negative health impacts of navigating the healthcare system for musculoskeletal conditions: A scoping review protocolPLOS ONE

Dear Dr. Eubank,

Thank you for submitting your manuscript to PLOS ONE. After careful consideration, we feel that it has merit but does not fully meet PLOS ONE’s publication criteria as it currently stands. Therefore, we invite you to submit a revised version of the manuscript that addresses the points raised during the review process.

We look forward to receiving your revised manuscript.

Kind regards,

Amin Nakhostin-Ansari

Academic Editor

PLOS ONE

Journal Requirements:

Reviewers' comments:

Reviewer's Responses to Questions

**Comments to the Author**

1. Does the manuscript provide a valid rationale for the proposed study, with clearly identified and justified research questions?

Reviewer #1: Yes

Reviewer #2: Yes

2. Is the protocol technically sound and planned in a manner that will lead to a meaningful outcome and allow testing the stated hypotheses?

Reviewer #1: Yes

Reviewer #2: Yes

3. Is the methodology feasible and described in sufficient detail to allow the work to be replicable?

Reviewer #1: Yes

Reviewer #2: Yes

4. Have the authors described where all data underlying the findings will be made available when the study is complete?

Reviewer #1: Yes

Reviewer #2: No

5. Is the manuscript presented in an intelligible fashion and written in standard English?

Reviewer #1: Yes

Reviewer #2: Yes

6. Review Comments to the Author

You may also provide optional suggestions and comments to authors that they might find helpful in planning their study.

Reviewer #1: Excellent concept and research question. The methodology has been well thought of. A couple of main things to address:

Settings was not well described and no sub-group analysis planned. Is this specific to some countires or region? The perception or consequences of navigating health system will be different for different countries. So analysis needs to be done with this lens.

The search strategy is not very comprehensisve and may not pick up many MSK issues. This needs looking into.

Reviewer #2: Thanks for the protocol. This looks a well thought out and detailed study protocol. Just a couple of things I wasn't sure about:

1. Is this study looking at qualitative or quantitative papers? I’m assuming both but it might be good to explicitly state this.

2. Do you have plans for regional i.e. country specific analysis? I’d be tempted to add location of cohort to your extraction sheet. You may find very interesting differences between different countries and regions due to differing social contexts.

3. The search strategy limits articles to adult populations, but I don’t think this is detailed in inclusion/exclusion criteria. What age group are you looking at?

4. Inclusion criteria 2nd para, models of health. Is this part of the inclusion criteria? It seems a little disjointed. If so I think the paper should state something like ‘the health impact described must fit within one of eight dimensions’ or ‘we define health impact as’

5. Will you be searching the ref lists of included papers? i.e. backwards and forwards searching. Will any papers be included by expert recommendation etc?

6. Are you including people seeking medical help for traumatic injury? Your paper seems written up in the context of longer term conditions but if a study looks at emergency admissions for knee trauma or sports injury would that be included?

7. How are you dealing with more generalised conditions, such as patient seeking help for general MSK pain (most of which will fall into knee, low back and shoulder)? Are these excluded, if so it may be useful to specify that papers exploring generalised conditions which may include knee, low back and shoulder are excluded)

8. The first sentence of the discussion seems overly complex, I think it may be better as two sentences, Also are resources narrowing everywhere? If it’s a global study that assumption may not be true everywhere.

This is a really interesting study and I look forward to reading the outcomes.

7. PLOS authors have the option to publish the peer review history of their article (what does this mean?). If published, this will include your full peer review and any attached files.

Reviewer #1: **Yes: **Opeyemi Babatunde

Reviewer #2: **Yes: **Martin J Stevens

---

## [Author Response · Author response to Decision Letter 0]

27 Aug 2024

TOPIC: Review response letter

MANUSCRIPT ID: PONE-D-24-26506

Dear Dr Amin Nakhostin-Ansari and Editorial Team: 

As per your instructions, received by email on August 15, 2024, this is our response letter accompanying the submission of manuscript (PONE-D-24-26506) to PLOS One. Below we provide a response to each point raised by the two reviewers from PLOS One, with reference to where changes can be viewed in the submitted manuscript. 

We thank the reviewers for their helpful review and comments and have detailed the changes made in the submitted manuscript.

Editor Comment 1: Please ensure that your manuscript meets PLOS ONE's style requirements, including those for file naming. The PLOS ONE style templates can be found at 

Author Response 1: We have adhered to the style requirements to the best of our knowledge.

Editor Comment 2: Please provide a complete Data Availability Statement in the submission form, ensuring you include all necessary access information or a reason for why you are unable to make your data freely accessible. If your research concerns only data provided within your submission, please write "All data are in the manuscript and/or supporting information files" as your Data Availability Statement.

Author Response 2: We have updated our data availability statement in the submission site to state:

"All data are in the manuscript and/or supporting information files"

Reviewer 1 Comment 1: Excellent concept and research question. The methodology has been well thought of. A couple of main things to address:

Author Response 1: We thank the reviewer for their encouraging comment and for taking the time to provide a review of our manuscript.

Reviewer 1 Comment 2: Settings was not well described and no sub-group analysis planned. Is this specific to some countires or region? The perception or consequences of navigating health system will be different for different countries. So analysis needs to be done with this lens.

Author Response 2: We have included a new category in our extraction template to chart each paper based on study setting (e.g., acute, primary, community) and location of study (e.g., region, country). These changes are located on page 15, Table 2, lines 152-153 and page 16, line 159.

Reviewer 1 Comment 3: The search strategy is not very comprehensisve and may not pick up many MSK issues. This needs looking into.

Author Response 3: Thank you for your comment and the opportunity to further explain our search strategy. We purposively selected the most common MSK conditions (low back, knee, shoulder) and those that typically require healthcare system intervention. Additionally, these three MSK conditions have been identified as priority conditions in Alberta, Canada as this scoping review will inform current initiatives of the MSK-Transformation Project of Alberta Health Services in Canada. We worked with a trained research librarian to craft the search strategy and piloted the strategy to ensure that exemplar papers we previously identified were captured in the search. Although our strategy will not capture all MSK conditions (e.g., hand, hip) we do believe it is comprehensive enough to capture papers examining the focal MSK conditions (i.e., knee, low back, shoulder).

We have revised the methods (Page 6-7, lines 115-119) and limitation section (Page 17, lines 189-192) to provide more details on the search strategy and acknowledge that our strategy is not intended to capture all MSK conditions.

Reviewer 2 Comment 1: Thanks for the protocol. This looks a well thought out and detailed study protocol.

Author Response 1: We thank the reviewer for their kind comment and thoughtful review.

Reviewer 2 Comment 2: Is this study looking at qualitative or quantitative papers? I’m assuming both but it might be good to explicitly state this.

Author Response 2: The reviewer is correct; this study is reviewing both qualitative and quantitative papers. We have clarified this point in text (Page 6, lines 102-103).

Reviewer 2 Comment 3: Do you have plans for regional i.e. country specific analysis? I’d be tempted to add location of cohort to your extraction sheet. You may find very interesting differences between different countries and regions due to differing social contexts.

Author Response 3: We have included a new category in our extraction template to chart each paper based on study setting (e.g., acute, primary, community) and location of study (e.g., region, country). This change is located on page 15, Table 2, lines 152-153 and page 16, line 159.

Reviewer 2 Comment 4: The search strategy limits articles to adult populations, but I don’t think this is detailed in inclusion/exclusion criteria. What age group are you looking at?

Author Response 4: We are looking at adult populations (i.e., 18 or older). We have revised the inclusion criteria to include this criterion (Page 5, lines 83-84).

Reviewer 2 Comment 5: Inclusion criteria 2nd para, models of health. Is this part of the inclusion criteria? It seems a little disjointed. If so I think the paper should state something like ‘the health impact described must fit within one of eight dimensions’ or ‘we define health impact as’

Author Response 5: We have revised the second paragraph to incorporate the reviewer’s suggested language (Page 5, lines 90-91).

Reviewer 2 Comment 6: Will you be searching the ref lists of included papers? i.e. backwards and forwards searching. Will any papers be included by expert recommendation etc?

Author Response 6: We will be searching the reference lists for included papers (i.e., citation chaining), we have now included this statement in the methods section (Page 13, lines 140-146).

Prior to connecting with the research librarian, our team (which included bone and joint health experts) assembled a small number of exemplary papers which met the inclusion criteria. We tested the accuracy of the search strategy by checking if the exemplary papers were included within the results from the search. If not, the strategy was revised. We have now included some text in the methods section explaining this additional step (Page 6-7, lines 115-119).

Reviewer 2 Comment 7: Are you including people seeking medical help for traumatic injury? Your paper seems written up in the context of longer term conditions but if a study looks at emergency admissions for knee trauma or sports injury would that be included?

Author Response 7: Yes, we are including both chronic and acute MSK conditions. We have revised the manuscript to include this detail (Page 6, line 109).

Reviewer 2 Comment 8: How are you dealing with more generalised conditions, such as patient seeking help for general MSK pain (most of which will fall into knee, low back and shoulder)? Are these excluded, if so it may be useful to specify that papers exploring generalised conditions which may include knee, low back and shoulder are excluded)

Author Response 8: Generalized MSK pain is included in our search strategy. The rationale for this is that MSK pain is typically the reason individuals will seek healthcare and thus have to navigate the healthcare system. Many of these individuals who seek help for pain do go on to be diagnosed with an MSK condition. We have revised the manuscript to include generalized pain as part of the search (Page 6, line 110-111).

Reviewer 2 Comment 9: The first sentence of the discussion seems overly complex, I think it may be better as two sentences, Also are resources narrowing everywhere? If it’s a global study that assumption may not be true everywhere.

Author Response 9: We have revised the first sentence of the discussion to split it into two sentences (Pages 16-17, lines 172-177). In addition, we have provided two citations to support the narrowing of healthcare services for chronic (including MSK) conditions globally. First, we cite a WHO press release that discusses the progress of one of the World Health Organization Sustainable Development Goals: the expansion of essential health services. In this article the authors report that since 2015 there have been no improvements in access to health services for non-communicable diseases and this is in the face of increasing global populations. Second, we cite an article examining unmet need for rehabilitation using data from the Global Burden of Disease project, this article finds that need for rehabilitations has increased by 63% from 1990 to 2019 internationally.

We again thank the editor and reviewers for their comments, and we hope these revisions are satisfactory.

---

## [Decision Letter · Decision Letter 1]

13 Sep 2024

PONE-D-24-26506R1Negative health impacts of navigating the healthcare system for musculoskeletal conditions: A scoping review protocolPLOS ONE

Dear Dr. Eubank,

Thank you for submitting your manuscript to PLOS ONE. After careful consideration, we feel that it has merit but does not fully meet PLOS ONE’s publication criteria as it currently stands. Therefore, we invite you to submit a revised version of the manuscript that addresses the points raised during the review process.

We look forward to receiving your revised manuscript.

Kind regards,

Amin Nakhostin-Ansari

Academic Editor

PLOS ONE

Journal Requirements:

Reviewers' comments:

Reviewer's Responses to Questions

**Comments to the Author**

1. Does the manuscript provide a valid rationale for the proposed study, with clearly identified and justified research questions?

Reviewer #2: Yes

Reviewer #3: Yes

2. Is the protocol technically sound and planned in a manner that will lead to a meaningful outcome and allow testing the stated hypotheses?

Reviewer #2: Yes

Reviewer #3: Yes

3. Is the methodology feasible and described in sufficient detail to allow the work to be replicable?

Reviewer #2: Yes

Reviewer #3: Yes

4. Have the authors described where all data underlying the findings will be made available when the study is complete?

Reviewer #2: Yes

Reviewer #3: Yes

5. Is the manuscript presented in an intelligible fashion and written in standard English?

Reviewer #2: Yes

Reviewer #3: Yes

6. Review Comments to the Author

You may also provide optional suggestions and comments to authors that they might find helpful in planning their study.

Reviewer #2: Thanks for your response which answers all my queries. I believe the changes and clarifications make the paper clearer. Good luck with the study

Reviewer #3: I have reviewed the manuscript with great interest and enthusiasm. The significance of the title is noteworthy, and the manuscript is presented in an appropriate format. It would benefit from minor revisions to further enhance its clarity and impact. Upon addressing these revisions, the manuscript appears well-suited for publication in PLOS ONE.

- In the eligibility criteria, the manuscript mentions the use of an eight-dimension health model to define health impacts. While this is a solid approach, it would benefit from a clear definition of what constitutes a health impact in each dimension (e.g., what qualifies as a financial or environmental health impact). Include specific examples or parameters that would be categorized under each dimension of health (social, emotional, physical, etc.) to improve replicability. For instance, would loss of employment due to prolonged pain fall under financial health impacts? Clarifying these boundaries will enhance the scoping review's focus.

- The manuscript emphasizes that no geographical limits will be applied to the search strategy. This global approach is commendable, but there should be consideration for regional variability in healthcare systems. Consider adding an analysis or categorization by country/region to assess if the negative health impacts vary by healthcare system structures or if not applicable, authors should discuss it in limitations.

- The manuscript's search strategy focuses on the three most common MSK conditions (low back, knee, shoulder). While this focus is understandable, it might limit the scope of the review as other MSK conditions, such as hip and neck issues, are also prevalent and burdensome.

- Patients with MSK conditions often have comorbidities (e.g., obesity, cardiovascular issues) that might complicate their navigation through the healthcare system. It could be beneficial if authors clarify how the review will handle studies where comorbidities are present.

7. PLOS authors have the option to publish the peer review history of their article (what does this mean?). If published, this will include your full peer review and any attached files.

Reviewer #2: **Yes: **Martin J. Stevens

Reviewer #3: **Yes: **Mohamad Mehdi Khadembashiri

---

## [Author Response · Author response to Decision Letter 1]

20 Sep 2024

TOPIC: Review response letter

MANUSCRIPT ID: PONE-D-24-26506R1

Dear Dr Amin Nakhostin-Ansari and Editorial Team: 

As per your instructions, received by email on September 13, 2024, this is our response letter accompanying the submission of manuscript (PONE-D-24-26506 R2) to PLOS One. Below we provide a response to each point raised by the two reviewers from PLOS One, with reference to where changes can be viewed in the submitted manuscript. 

We thank the reviewers for their helpful review and comments and have detailed the changes made in the submitted manuscript.

Reviewer 2 Comment 1: Thanks for your response which answers all my queries. I believe the changes and clarifications make the paper clearer. Good luck with the study

Author Response 1: We thank the reviewer for their response and for taking the time to provide a review of our manuscript. Their suggestions contributed to a much stronger manuscript.

Reviewer 3 Comment 1: I have reviewed the manuscript with great interest and enthusiasm. The significance of the title is noteworthy, and the manuscript is presented in an appropriate format. It would benefit from minor revisions to further enhance its clarity and impact. Upon addressing these revisions, the manuscript appears well-suited for publication in PLOS ONE.

Author Response 1: We thank the reviewer for their encouraging comment and for taking the time to review our manuscript. 

Reviewer 3 Comment 2: In the eligibility criteria, the manuscript mentions the use of an eight-dimension health model to define health impacts. While this is a solid approach, it would benefit from a clear definition of what constitutes a health impact in each dimension (e.g., what qualifies as a financial or environmental health impact). Include specific examples or parameters that would be categorized under each dimension of health (social, emotional, physical, etc.) to improve replicability. For instance, would loss of employment due to prolonged pain fall under financial health impacts? Clarifying these boundaries will enhance the scoping review's focus.

Author Response 2: Thank you for your comment and the clarify our chosen model of health. We have revised the manuscript to include definitions and examples of each of the eight dimensions of the health model in the newly include Table 1 on Page 5-6. Of note, upon reviewing our referred sources again, we opted to collapse financial health into a large group of “occupational health”. The reviewer’s example of loss of employment due to prolonged pain due to untreated MSK conditions would fall under occupational health impacts.

Reviewer 3 Comment 3: The manuscript emphasizes that no geographical limits will be applied to the search strategy. This global approach is commendable, but there should be consideration for regional variability in healthcare systems. Consider adding an analysis or categorization by country/region to assess if the negative health impacts vary by healthcare system structures or if not applicable, authors should discuss it in limitations.

Author Response 3: Thank you for this suggestion. We had previously included country/region in our extraction criteria. We have now revised our extraction criteria to include type of healthcare system (e.g., universal healthcare system, social health insurance, private health insurance, out-of-pocket system, mixed system, community-based or faith-based healthcare systems) (see Table 3, formerly Table 2 on page 15). In addition, we have clarified in our methods section our intention to conduct a stratified synthesis by type of health system to examine any differences in negative health impacts by healthcare system structure. This sentence on page 17 lines 170-173 now states, “In addition, given that we have not placed any geographic limits on this scoping review, we will examine whether there are any differences in negative health impacts, or causes of impacts, based on the type of healthcare system in place in the study setting.”

Reviewer 3 Comment 4: The manuscript's search strategy focuses on the three most common MSK conditions (low back, knee, shoulder). While this focus is understandable, it might limit the scope of the review as other MSK conditions, such as hip and neck issues, are also prevalent and burdensome.

Author Response 4: The reviewer is correct; this study is limited in its examination of conditions such as hip and neck issues. We purposively selected three of the most common MSK conditions (low back, knee, shoulder) and those that typically require healthcare system intervention. Additionally, these three MSK conditions have been identified as priority conditions in Alberta, Canada as this scoping review will inform current initiatives for the healthcare system in Alberta, Canada. Although our strategy will not capture all MSK conditions we do believe it is comprehensive enough to capture papers examining the focal MSK conditions (i.e., knee, low back, shoulder). We have recognized this limitation on page 18 lines 191-194. 

Reviewer 3 Comment 5: Patients with MSK conditions often have comorbidities (e.g., obesity, cardiovascular issues) that might complicate their navigation through the healthcare system. It could be beneficial if authors clarify how the review will handle studies where comorbidities are present.

Author Response 5: The reviewer is correct, patients with MSK conditions often have comorbidities when seeking healthcare, therefore we have included an additional column in our extraction criteria on patient characteristics where these details can be extracted and summarized to provide important contextual information in the synthesis. We have revised the extraction table (Table 3, formerly Table 2 on page 15).

We again thank the editor and reviewers for their comments, and we hope these revisions are satisfactory.

---

## [Editor Report · Decision Letter 2]

30 Sep 2024

Negative health impacts of navigating the healthcare system for musculoskeletal conditions: A scoping review protocol

PONE-D-24-26506R2

Dear Dr. Eubank,

We’re pleased to inform you that your manuscript has been judged scientifically suitable for publication and will be formally accepted for publication once it meets all outstanding technical requirements.

Kind regards,

Amin Nakhostin-Ansari

Academic Editor

PLOS ONE
---

## [Editor Report · Acceptance letter]

16 Oct 2024

PONE-D-24-26506R2 

PLOS ONE

Dear Dr. Eubank, 

I'm pleased to inform you that your manuscript has been deemed suitable for publication in PLOS ONE. Congratulations! Your manuscript is now being handed over to our production team.

Kind regards, 

on behalf of

Dr. Amin Nakhostin-Ansari 

Academic Editor

PLOS ONE